# Machine Learning-Based Estimation of Ground Reaction Forces and Knee Joint Kinetics from Inertial Sensors While Performing a Vertical Drop Jump

**DOI:** 10.3390/s21227709

**Published:** 2021-11-19

**Authors:** Serena Cerfoglio, Manuela Galli, Marco Tarabini, Filippo Bertozzi, Chiarella Sforza, Matteo Zago

**Affiliations:** 1Dipartimento di Elettronica, Informazione e Bioingegneria, Politecnico di Milano, 20133 Milano, Italy; serena.cerfoglio@polimi.it (S.C.); manuela.galli@polimi.it (M.G.); 2E4Sport Laboratory, Politecnico di Milano, 23900 Lecco, Italy; marco.tarabini@polimi.it; 3Dipartimento di Meccanica, Politecnico di Milano, 20133 Milano, Italy; 4Department of Biomedical Sciences for Health, Università degli Studi di Milano, 20133 Milano, Italy; filippo.bertozzi@unimi.it (F.B.); chiarella.sforza@unimi.it (C.S.)

**Keywords:** vertical drop jump, wearable sensors, neural networks

## Abstract

Nowadays, the use of wearable inertial-based systems together with machine learning methods opens new pathways to assess athletes’ performance. In this paper, we developed a neural network-based approach for the estimation of the Ground Reaction Forces (GRFs) and the three-dimensional knee joint moments during the first landing phase of the Vertical Drop Jump. Data were simultaneously recorded from three commercial inertial units and an optoelectronic system during the execution of 112 jumps performed by 11 healthy participants. Data were processed and sorted to obtain a time-matched dataset, and a non-linear autoregressive with external input neural network was implemented in Matlab. The network was trained through a train-test split technique, and performance was evaluated in terms of Root Mean Square Error (RMSE). The network was able to estimate the time course of GRFs and joint moments with a mean RMSE of 0.02 N/kg and 0.04 N·m/kg, respectively. Despite the comparatively restricted data set and slight boundary errors, the results supported the use of the developed method to estimate joint kinetics, opening a new perspective for the development of an in-field analysis method.

## 1. Introduction

The Vertical Drop Jump (VDJ) is a plyometric training exercise commonly used by coaches to develop a wide variety of athletic biomotor qualities, such as speed strength, sprinting speed, and explosive power of the lower limb [1]. The VDJ requires the athlete to step from a measured drop height and, after landing on the ground with a limited leg flexion, to perform a maximal vertical jump, with a short ground-contact period [1,2].

The VDJ is usually part of a training program to enhance performance and neuromuscular readiness, and it has been widely used as clinical test for injury-risk screening purpose for Anterior Cruciate Ligament (ACL) non-contact injury assessment and evaluation of training interventions [1,3,4,5], while the evidence of its capability of predicting future injuries is still scarce [6,7,8,9,10,11].

Biomechanical factors and specific at-risk movement patterns have been assessed in the VDJ with complex 3D motion analysis set-ups based on optical motion capture systems. So far, such systems are the gold standard to evaluate kinematics and kinetics of lower extremity in a controlled laboratory setting, performed in order to replicate postures consistent with real at-risk sporting situations [7,12,13,14].

The set of kinematic and kinetic variables obtained by using 3-D systems enables the evaluation of lower extremity musculoskeletal loading and of at-risk multi-planar knee movements for screening purposes [9,11,15] leading to significant advances in the understanding of landing biomechanics.

However, these methods are not easy to apply as a large-scale screening tool: they are time-consuming, expensive, and their measurements are potentially affected by methodological issues (e.g., errors while identifying anatomical landmarks) or skin movement artefacts [7,12,15].

Additionally, the athlete is required to move into to a restricted area of action (close to a force plate and/or an instrumented laboratory) while wearing the equipment for acquisition (markers), and those restrictions may influence the movement pattern.

All these considerations indicate that there is a need for a simple and accurate method to capture knee kinematics and kinetics to assess the risk of injury during jumping tasks.

The use of wearable inertial-based systems using Inertial Measurement Units (IMUs) is a promising alternative to the use of the gold standard optoelectronic systems [12,16,17,18]. IMU-based systems do not require the use of cameras or complex laboratory settings to perform the test. Inertial sensors can be easily attached with straps to the limbs of the athlete and addressed separately to define body parts, allowing for a direct determination of kinematic parameters, providing data of angular acceleration and velocity, as well as spatial orientation of the individual body segments [12,19].

However, it should be noticed that, similar to the optical markers, IMUs should be placed in specific bony landmarks and not directly on the muscle to reduce soft tissue artefacts caused by the displacement of the sensor of the underlying tissues with respect to the bone [17,18,20].

IMUs are designed to be small, lightweight, and wireless, thus permitting full movements while participating in a sport or a test, and they do not require a predefined area of action to which the movement must be restricted. As the athlete is free to move, the collected data are potentially more consistent with a real sport situation (i.e., increased ecological validity of testing) and can be used to enhance athletic performance as well as to identify at-risk motion patterns [12,21,22].

IMU-based systems have already been successfully used to detect kinematic parameters in track and field applications [21] as well as in gait analysis [20,23,24].

IMU-based systems offer the possibility to directly assess temporal, dynamic, and kinematic parameters but IMUs raw data *per se* cannot be used as input parameter for inverse dynamic calculations. Different approaches based on machine learning methods and Artificial Neural Networks (ANNs) are used to support human motion analysis to assess kinetic parameters such as GRFs and joint moments [25,26,27].

ANNs are inspired by the structure of the human brain and serve as universal approximator, able to realise an arbitrary mapping of one vector space onto another vector space [23,28]. Instead of explicitly programming the solution of one specific task, ANNs are able to capture and use some a priori unknown information hidden in given data (training dataset), learn from them, and apply the learned input–output relationship to new data, allowing for a prediction of the behaviour of a system with respect to previously unseen data [19,23,29].

Many studies addressed the usability of ANNs to estimate ground reaction forces or joint moments by means of kinematic data obtained from an optical motion analysis system [26] or where such methods enable the use of IMU sensors for the measurement and, simultaneously, optical data for training purpose [23]. Most of the existing research uses optical motion capture systems to apply kinetics prediction methods to kinematic data. Advancements of the combined use of IMU data and ANNs, when properly validated [30], might have the potential to overcome the actual limitations of a time-consuming traditional laboratory set-up [12,25,31].

Various studies [5,9,10,32] have estimated knee joint load during jumping tasks such as the VDJ, but no study has directly assessed knee joint load using wearable sensors so far. In particular, we have no evidence of ANN-based approaches to predict knee joint kinetics during the landing phase. Thus, the aim of this study was to develop and validate an ANN allowing for the estimation of knee joint moments and GRFs during the Vertical Drop Jump by using IMUs data.

The findings of this study could help to develop a portable, practical, and field-based method to quantify and monitor knee kinetics. IMU-instrumented assessment would allow us to easily collect large datasets of objective biomechanical data in order to monitor an athlete’s workload, to set up effective training programs [26], as well as to identify at-risk behaviors that may result in a ligamentous injury.

## 2. Materials and Methods

### 2.1. Participants and Experimental Procedures

The study involved the simultaneous collection of data with commercial IMU sensors (Physilog^®^ by Gait Up Ltd., Lausanne, Switzerland), an optoelectronic marker-based motion capture system (SMART DX 400 system by BTS Bioengineering SPA, Milan, Italy), and a force plate (AMTI OR6-7 force platform by Advanced Mechanical Technology, Inc., Watertown, MA, USA) during the execution of the Vertical Drop Jump.

Eleven healthy subjects (five women, six men), aged between 23 and 29 years, were recruited on a voluntary basis and asked to perform up to 10 VDJs in a row. Inclusion criteria were not being under- or overweighted (body mass index between 18.5 and 24.9) and the absence of any knee-injury history. Of the considered cohort, three subjects were regular amateur sports practitioners, whilst the other were students with an active lifestyle, occasionally attending the gym and who walk for daily commuting.

Before the tests, each subject gave written informed consent to participate in the study. Subjects’ anthropometric data are reported in Table 1.

Each session began with a short warm up after which the subjects were instructed to drop off a 30 cm box and perform a maximal VDJ upon landing.

Participants were asked to stand upright on the box and instructed to start the movement by stepping out from the box with a single leg as well as to place arms on the hips throughout the execution of the jump, according to the reference technical execution model [1]. All subjects were previously acquainted with the experimental tasks and were able to cope with the provided instructions and to complete the test.

### 2.2. Experimental Setup

The stereophotogrammetric system used for the optical data acquisition consisted of a SMART DX 400 system (BTS Bioengineering SPA, Milan, Italy) with a sampling frequency of 100 Hz. The system was calibrated prior to each usage according to the manufacturer’s specification.

The laboratory was also equipped with two 46.5 × 51.8 cm^2^ AMTI OR6-7 force platforms (Advanced Mechanical Technology, Inc., Watertown, MA, USA), sampling at 200 Hz, to collect Ground Reaction Force (GRF) data. The contact phase of the first landing of each VDJ was defined as the period where the unfiltered vertical GRF exceeded 10 N. Each VDJ was considered successful if the subject landed with one foot on each of the two force plates and all the reflective markers were visible by the cameras throughout the whole jump. Optical and ground reaction forces data were automatically synchronized.

#### 2.2.1. Marker Set

Reflective skin markers (Ø = 10–15 mm) coated with an aluminum powder film and mounted on a plastic tip were attached over anatomical landmarks on the legs and pelvis of each subject according to a lower-body model [33] using 30 markers (Figure 1).

Landmarks’ position was manually determined by palpation, by identifying regions with reduced thickness of the tissues interposed between bone and skin. According to this approach, 18 markers were bilaterally placed on the superior-anterior iliac spine, posterior iliac crest, lateral and medial femoral epicondyle, lateral and medial malleolus, calcaneus and foot (corresponding to the 1st and 5th metatarsal heads).

Furthermore, 12 markers fixed at four clusters were attached to subject’s thigh and shank on both legs. Each cluster included three markers located at a fixed distance, and it defines a technical reference system fixed to the segment to which it is attached. The anthropometric model was composed of rigid segments, representing the bony segments of the subject, and joint kinematics was defined as the relative orientation in time between two adjacent segments [34].

#### 2.2.2. IMU Devices

Three wearable IMU devices (Physilog^®^ by Gait Up Ltd., Lausanne, Switzerland) were used to measure and acquire the VDJs. Sensors were configured via the Research ToolKit Desktop software and organized into a multiple sensor network, including a single master sensor and two slave sensors to record synchronous signals.

The IMUs synchronized regularly during the measurement, and they were easily turned on/off wirelessly through the provided mobile application (GaitUp companion Android application. Lausanne, Switzerland). Raw data were stored on the onboard SD memory of each sensor and then transferred to a PC for offline analysis. The slave sensors were placed via Velcro^®^ straps on the participant’s right thigh and shank, whilst the master sensor was placed with a rubber clip on the sacrum, between the posterior and superior iliac spine.

Each sensor is a six-axes stand-alone unit integrating a micro-controller, an internal memory, a battery (Lithium-Ion Polymer Accumulator, 3.7 V 140 mAh by Renata, Itingen, Switzerland), a barometric pressure sensor, and a 3D accelerometer and a 3D gyroscope with programmable ranges. Device settings were set to a sampling frequency of 128 Hz and a dynamic range of ±16 g for the accelerometer and ±2000 degrees/s for the gyroscope.

Each unit is a compact (47.5 mm × 26.5 mm × 10 mm) and lightweight (36 g) wearable device for sensing movement, allowing for a long-term motion recording and on-board processing. Physilog^®^ does not require any calibration before the acquisition because this are estimated during the recording and its right-handed coordinate system is oriented as displayed in Figure 2, together with the locations of the sensors.

An optical maker was placed on the master sensor and used as a synchronization reference for the data recorded by the motion capture system and the IMUs.

Prior to each jumping session, an operator moved the master sensor (and the attached marker therein) up and down five or more times to obtain a quasi-periodical signal recognizable from both the IMU and optical system records.

### 2.3. Data Analysis and Processing

A total of 112 successful VDJs were recorded, and different software applications were used for complete data processing and analysis. The optical raw data were processed with SmartTracker (BTS Bioengineering, Milan, Italy) and then exported in c3d format for further processing in Visual3D (version 6.03.06, C-Motion, Inc. Germantown, MD, USA). Visual3D makes use of the relative positions of at least three non-aligned markers to define body segment as a rigid body, described by scaled inertial properties [32]. Such a model (Figure 3) was first fit to the static recording of each subject. Then, the calibrated model was matched to the drop jump records to calculate the joint angles (kinematics) and then the joint moments (kinetics). In particular, joint kinetics was obtained by applying the inverse dynamics analysis to the kinematics of the biomechanical model and to the location, magnitude, and direction of the ground reaction forces acting on the foot during the first landing phase of the VDJ (externally applied forces).

Joint moments were computed according to this approach for the hip, knee, and ankle. In the study, only the right knee joints moments were selected and used for further analysis. The conventions used to represent the moments are reported in Table 2.

Additionally, Visual3D allowed for event detection. In particular, the first landing phase of each jump (in the following, landing phase) was defined as the interval between the time instant in which the feet were in contact with the ground, when the vertical GRF exceeded the 10-N threshold, and the moment in which the feet lifted from the ground to perform the jump. Data were exported to Matlab (version R2020b, The MathWorks Inc, Natick, MA, USA) for further analysis with custom scripts.

The raw data simultaneously collected by the IMUs were transferred via USB to the computer and then accessed through Research ToolKit (RTK) Desktop software (Gait Up Ltd., Lausanne, Switzerland). An example of a signal displayed by the sensor placed on the sacrum is shown in Figure 4.

For all the IMU sensors, the data collected in the same measurement were automatically assembled in a cluster and synchronized. Physilog^®^ 5 data were also exported to Matlab for further analysis.

### 2.4. Data Synchronization and Dataset Structure

To use the data acquired by the IMUs as input for the ANN, signals must be time-synchronized with those acquired by the marker-based motion capture system.

In this study, the synchronization trigger was the initial periodic movement of the master sensor attached to a reflective marker.

The synchronization was performed within Matlab. First, the vertical marker coordinate (*y*-axis) with respect to the global reference system of the laboratory, acquired at 100 Hz, was resampled to 128 Hz. The signal acquired by the master sensor along the *z*-axis of its own reference system was processed by subtracting the gravity component and by filtering it using a standard fourth order low-pass Butterworth filter with a 10-Hz cut frequency, according to [35]. Signals were manually cut to isolate a window containing the spikes to be matched and used as reference for the synchronization. The cross-correlation of the cut signals was calculated, and the resultant delay was used to align the full signals, as well as all the signals recorded by IMU units (Figure 5).

After synchronization, the 3-axes signals of right knee moment and right GRFs extracted from Visual3D were cut to isolate their values during the first landing phase of the VDJ using the event detection information obtained via Visual3D and resampled to 128 Hz (Figure 6).

Prior to cutting, IMU data had to be filtered to remove noise and to preserve the meaningful components of the signals by applying a standard fourth order low-pass Butterworth filter with a 32-Hz cut frequency. The aligned accelerometer and gyroscope signals of the three inertial units were cut in the same way as the signals recorded by the optoelectronic system (i.e., in correspondence with the landing phase). For each subject, a 25-column table containing the matched signals of both systems in the first landing phase of each drop jump, as well as a reference column with the indexes of the frames of each jump, was obtained (Figure 7).

The full data table was created by concatenating the tables of each subject.

Additionally, two other tables were generated in the same way using the IMU signals filtered with a cut frequency of 12 Hz and 24 Hz to evaluate the sensitivity of the outcome to the filter settings.

### 2.5. Neural Network Implementation

In the current research, a non-linear autoregressive network with an external input (NARX) neural network was implemented in Matlab. The NARX is a recurrent dynamic neural network with feedback connections enclosing several layers of the network. The NARX model is a nonlinear generalization of the Autoregressive Exogenous (ARX) model that is used as a standard instrument in black-box system identification [36] and is commonly used in time series modelling in an extensive variety of dynamic systems [37].

A standard feedforward NARX network can predict the output response through regression of past output and exogenous input values [38].

The NARX model output is defined by the following equation:(1)$$y\left(t\right)=f\left(y\left(t-1\right),y\left(t-2\right),\;\ldots ,\;y\left(t-n_{y}\right),\;u\left(t-1\right),u\left(t-2\right),\ldots ,u\left(t-n_{u}\right)\right)$$
where f(.) is the mapping function of the *y(t)* output response that is regressed by the previous time series values and past values of input data (*u*) and $n_{y}$ and $n_{u}$ are the time delays. This implementation allows for a vector ARX model, where the input and the output can be multidimensional.

The NARX neural network model allows for two different architectures, shown in Figure 8. The first is called series–parallel architecture (SP mode, open-loop), where the output $\hat{y}\left(t\right)$ is computed by using only the actual values of both endogenous (*u*(*t*)) and exogenous inputs (*y*(*t*)). The second one is the parallel architecture (closed loop), where the output $\hat{y}\left(t\right)$ is computed by using the actual values of the endogenous inputs $u\left(t\right)$ and the past predicted values of the estimated output $\hat{y}\left(t\right)$ that are fed back.

In this study, the series–parallel architecture was used as the true values of the input series (the data collected by the IMU units), and the true outputs (the GRFs and the right knee joint moments) were always available during the training of the network.

The training of the neural network was accomplished through a train–test technique. The data contained in the data matrix, which contains known input and output values, was split into a training and a test set: the VDJs performed by seven of the 11 subjects were randomly chosen and used as the training set whilst the data of the remaining four subjects served as the test set.

Data recorded by the IMUs were given as input, and the optical motion capture data served as output (target). Input and output details are reported in Table 3.

The NARX was implemented via the in-built function narxnet of the Matlab neural network toolbox. The ANN topology was determined through a trial-and-error procedure, starting from a set of hyperparameters (i.e., number of hidden layers, number of neurons therein, and so on) that could be retrieved from similar existing research (MUNDT). Then, such parameters were adjusted iteratively based on the network’s output metrics.

The input and feedback delays were set as default, ranging from 1:2, while the number of hidden layers and neurons was systematically optimized via a trial-and-error process in order to adapt the capacity of the network.

A delayed version of the input time series data was prepared using the function *preparets* that automatically shifted the input and target series to as many steps as needed to fill the initial input and layer delay states. These input and layer states were then employed as inputs to MATLAB’s built-in neural network training function [39].

Weights and biases were randomly set whilst configuring the network on data, and regularization was introduced to prevent overfitting and to improve the network’s ability to generalize by modifying the in-built Matlab default performance function. The regularization factor was set to 0.5.

The training dataset was used to train the neural network with a back-propagation algorithm. Different back-propagation algorithms were tested (e.g., Resilient backpropagation and the Levenberg–Marquadt algorithm) together with different activation functions for each layer, belonging to the sigmoid (hyperbolic tangent sigmoid and logarithmic sigmoid transfer) and linear (positive linear and linear transfer) function families.

During the training process, the results produced for the training set by the model were compared with the real value for each output vector fed to the net. According to this comparison and the specific learning algorithms, weights and biases were tuned to optimize the network performance function, defined as the regularized mean square error (MSE) between the network output and the target output, so that the computed values could match the known real output values.

After the training phase, the network was applied to the test set.

The network was cross validated (20-fold cross validation): the full dataset was iteratively split into different training (*n* = 7 subjects) and test (*n* = 4 subjects) sets in order to obtain insight into the network’s ability to generalize to an independent dataset and to predict new data that were not used in the training phase.

The network’s performance was evaluated as the resulting average Root Mean Square Error (RMSE), a standard statistical metric used to measure the error of the model in predicting quantitative data. Formally, RMSE is defined as
(2)$$RMSE=\;\sqrt{\overset{n}{\underset{i=1}{\sum }}\frac{{\left(\hat{y_{i}}-y_{i}\right)}^{2}}{n}}=\;\sqrt{\overset{n}{\underset{i=1}{\sum }}\frac{e_{i}^{2}}{n}}$$
where $\hat{y_{1}}$...$\hat{y_{n}}$ are the predicted values, $y_{i}$…$y_{n}$ are the real values (thus, $e_{1}$…$e_{n}$ are the errors), and *n* is the number of observations.

The underlying assumption when presenting the RMSE is that the errors are unbiased and follow a normal distribution to provide a complete picture of the error distribution [40] RMSE was calculated for each predicted target.

Additionally, the error histogram containing the errors between target values and predicted values after training the network was obtained. The histogram indicates how predicted values differ from the target and it was calculated for both the training and test set.

The total error range on the *x*-axis was divided into 20 smaller bins, and the number of samples from the dataset in a particular bin are reported on the *y*-axis. Positive and negative error bins, accounting for the direction of the bias, were separated by a vertical orange line, corresponding to zero error.

The closer the predicted values to the target, the better the model performance.

## 3. Results

The topology of the developed NARX Neural Network is shown in Figure 9.

The input layer consisted of two 1:2 Tap Delay Lines that took the multi-dimensional endogenous time series *x*(t) (IMUs data) and the exogenous time series *y*(t) (right GRFs and right knee moments) and transmitted them in their time-delayed form into the subsequent layers. The computational engine of the network was made by two fully-connected hidden layers, composed of 1000 neurons each, where a linear activation function defined how the weighted sum of the input was transformed into the output from the nodes, finally yielding to the outputs of the whole network through the output layer.

Weights and biases were randomly initialized in each periodic training phase, and different backpropagation algorithms were tested to achieve the result.

Finally, the Resilient Backpropagation algorithm was chosen to train the final NARX network. The number of epochs was optimized via a trial-and-error process, and it was finally set to 5000, and the training stopped when the last epoch was reached.

The network was able to compute target values that closely matched the known real output values of the test set.

An example of the predicted patterns of the 3D right GRFs and the right knee moments obtained for the VDJs performed by one of the subjects chosen as the test set are reported in Figure 10, compared to their real (measured) values.

The predicted curves followed nearly the same pattern as the real curves, except at the very first contact with the ground, corresponding to the first frames of each landing phase of the VDJs.

The mean RMSE values achieved for the prediction of GRFs and right knee moments by the developed neural network are reported in Table 4.

Two additional datasets with the same structure of the main dataset were obtained by using a 12-Hz and a 24-Hz cut off frequency for the fourth order Butterworth filter applied to IMU data. The network was then trained by using those two datasets to see how the noise could affect the network’s ability to predict the data. However, the cut-off frequency slightly influenced the ability of the network to learn from the data.

The error histogram between target and predicted values, indicating how the predictions differed from the targets, was also obtained for both the training and test set. Two error histograms were obtained, one displaying the errors on the GRFs and the other displaying the errors on the right knee moments (Figure 11).

Most of the samples from the dataset were within the bins that were very close to the zero-error bar, indicating that the network was able to learn from the training set and then apply the learned pattern to the test set.

## 4. Discussion

In this paper, we showed that it was possible to estimate articular moments and GRFs during the first landing phase of a VDJ with an average RMSE of 0.02 N/kg and 0.04 N·m/kg by exploiting the acceleration and angular velocity recorded via inertial sensors combined with a neural network-based machine learning method.

### 4.1. Novelty of the Study

The predicted knee joint moments and GRFs showed a strong agreement with their measured values, according to their RMSE. Generally, there is no fixed threshold limit for RMSE but besides a small error, it would be optimal if the RMSE calculated on the training set was similar to the RMSE calculated on the test set to prevent overfitting (RMSE train > RMSE test) or underfitting (RMSE train < RMSE test).

The RMSE values were comparable for all the predicted variables, and they all respected the previously mentioned constraints, accounting for the capability of the network to predict kinetic data with a narrow error with respect to the articular moments and the GRFs acquired with traditional motion analysis methods.

A strength of the developed method lies in the ability of the network to simultaneously predict six output waveforms without the need of different models to estimate the two groups of kinetic variables during the execution of the same task.

Particular relevance was given here to the estimation of the 3D GRFs directly from the data recorded by inertial sensors, similar to previous work [41] in which the authors focused on the estimation of the vertical GRF (vGRF) during running using data from three body-worn IMUs by feeding them into two concatenated ANNs, where the first one mapped the relative orientations (expressed as quaternions) of the lower limbs to joint angles whilst the second one mapped the estimated joint angles in combination with vertical sensor acceleration with respect to vGRF.

Their model returned excellent results in matching the actual force profiles measured by the force plate, estimating the vGRF with an accuracy < 0.27 BW in terms of RMSE, which is in line with the results achieved in this study (RMSE vertical GRF = 0.04 N/Kg). However, it should be noticed that Wouda’s model [41] was developed to estimate the vertical GRFs during running, a highly dynamic and cyclic action in which the GRFs vary rapidly and therefore the information of the joint angles need to be included to achieve a better prediction of the kinetics.

The Physilog^®^ sensors used in this study allowed for the extraction of quaternions; however, as the analysis of the GRFs was conducted on a small temporal window referring to the first landing phase of the VDJ, it was decided not to include them within the model. It could be expected though that a deep network is able to incorporate sensor orientation, whose information is available as raw data.

The prediction performance of the knee joint moments was consistent with that achieved by Stetter et al. in 2020 [42], in which the authors developed an ANN able to estimate knee flexion and adduction moment during various locomotion tasks based on data obtained by two wearable inertial sensors located on the right thigh and shank. The moments estimated by the network showed a good agreement with their inverse dynamics-calculated counterpart, despite varying between tasks. In particular, the corresponding RMSE values obtained in this study for the knee FE moment and knee AA moment were 0.07 N·Kg/m and 0.03 N·Kg/m, respectively, lying almost in the same range of those reported in Table 4.

It should be noticed that task-specific modelling may lead to an increased estimation accuracy for individual tasks, and in turn to a lower RMSE with respect to a multi-tasking model, but it has the disadvantage that each task must be modelled by itself [42]. Following this observation, the network developed in this study was able to provide an accurate estimation for the VDJ kinetics, but it may not be suitable to make accurate predictions for data from other sporting tasks due to its high specificity for the current application.

The overall results of the study also confirmed the hypothesis of Mundt et al. (2019) [12] concerning the applicability of a feed-forward (FF) neural network to predict the joint moments and GRFs during sport-specific movements based on data from inertial sensors. The network developed hereby was not a standard FF network but a NARX network able to predict the output response through the regression of past outputs and exogenous input values. However, the use of the series–parallel architecture of the NARX allowed for the obtainment of a network exploiting the advantage of having two different series of input values, resulting in a more accurate prediction on the outputs, together with a pure feed-forward mechanism, as previously stated by Mundt et al. (2019) [12].

The network was developed on a dataset in which the duration in terms of frames of the first landing phase of each drop jump was not normalized and it showed a high variability according to the jumping behaviour of the subjects. As the network was able to learn the jumping pattern of each subject starting from non-time normalized series, the information about the individual jumping behaviour of each subject during the first landing phase of the drop jump was preserved, allowing for its potential use as indicator of the power output of the athlete [1].

### 4.2. Limitations

The network was able to learn from the data during the training phase and to then apply the learned pattern to new unseen data; however, it should be noticed that the network was not always able to make a sufficiently accurate prediction of the knee moments of the GRFs at the very first instant of the landing phase of the VDJ, corresponding to the first 2–3 frames of each jump (boundary artefacts). Prediction errors accumulated at the beginning of the curve but then the predicted joint moment followed nearly the same pattern as the measured waveform (Figure 12).

Despite this limitation, the ANN was able to give an accurate prediction of the values of the curves throughout the remaining duration of the landing phase of each jump. As the actual shape and peak of the varo/valgus moment was preserved, it potentially allowed for their use to make an evaluation of the landing behaviour for screening purposes for ACL injury.

Additional limitations should be considered. The present method focused on the estimation of the kinetic of the limb where two of the three IMUs were attached, without exploring the possibility of simultaneously predicting the same variables on the contralateral limb. Additionally, at present, the developed model did not allow for an online analysis of the kinetic variables during the execution of the drop jump and it still requires the use of data from force plates.

In fact, the detection of the events defining the beginning and the end of each landing phase was not implemented by using data directly from the IMUs, such as in Zago et al., (2021) [43], but it needed to be done via the data recorded by the force plates and processed with Visual3D (or an equivalent biomechanics software) afterwards.

The last concern was about the relatively restricted dataset used to train the network. The study was conducted during a global pandemic, and due to the limitation of the access to the laboratory and to the health protocols, only eleven subjects could take part to the trials. The performance of a machine learning method adaptively increases with the amount of data available for training, allowing the network to learn more motion patterns to improve its performance regarding new data [12,42]. Thus, the results achieved on a small dataset may not be extended to a wider dataset as the jumping behaviour is highly variable between subjects.

## 5. Conclusions

The current study investigated the possibility of an ANN-based method to predict knee joint kinetics and GRFs during the first landing phase of a VDJ, based on data recorded by three inertial sensors.

Despite the comparatively restricted dataset and the accumulation of boundary errors at the very beginning of the landing phase, the results supported the use of wearable inertial sensors in combination with machine learning-based methods to estimate joint kinetics in a sports application as the VDJ, highlighting the great potential of such methods. This approach may allow for the replacement of time-consuming inverse dynamics calculation, for trials with less equipment, time, and effort, as well as being feasible to test subjects in a natural environment and not only in a laboratory set-up [12,16].

Looking ahead, it should be investigated if a more complex architecture of the network may lead to an improvement in the prediction accuracy at the beginning of the landing phase, allowing for a consistent learning of the motion pattern along the curve and thus overcoming the error accumulation previously discussed.

Further improvement should be implemented to incorporate other targets into the prediction, e.g., hip adduction/abduction moment, which has been shown to have a key role in the ACL injury mechanism. The final goal would be to embed such neural networks into a mobile application controlling the sensors to display the moments at the knee joint and the GRFs in real time.

The method described in this study represents a promising starting point for further improvements leading to new pathways for in-field diagnosis methods that in turn could be applied to the early identification of at-risk landing behaviours and potentially highlight a link between injury risk and IMU-based instrumented assessment.

## Figures and Tables

**Figure 1 sensors-21-07709-f001:**
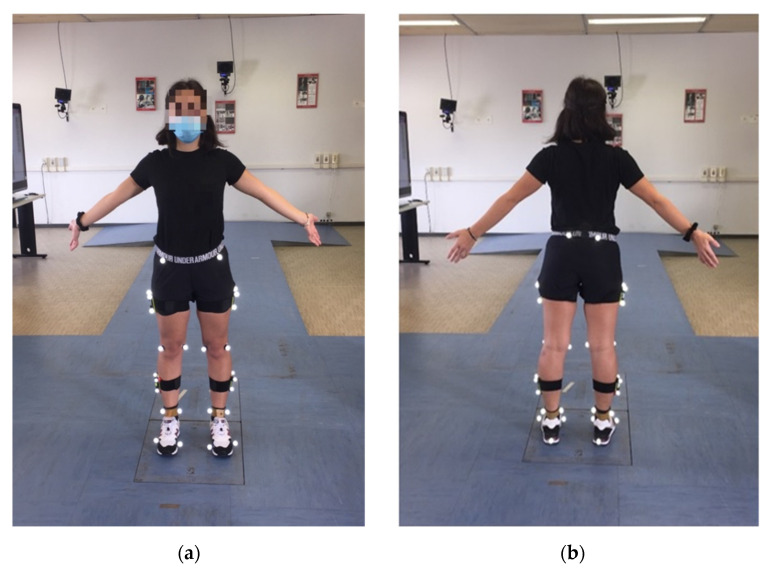
Subject equipped with the marker set used in the study: (**a**) front and (**b**) back view.

**Figure 2 sensors-21-07709-f002:**
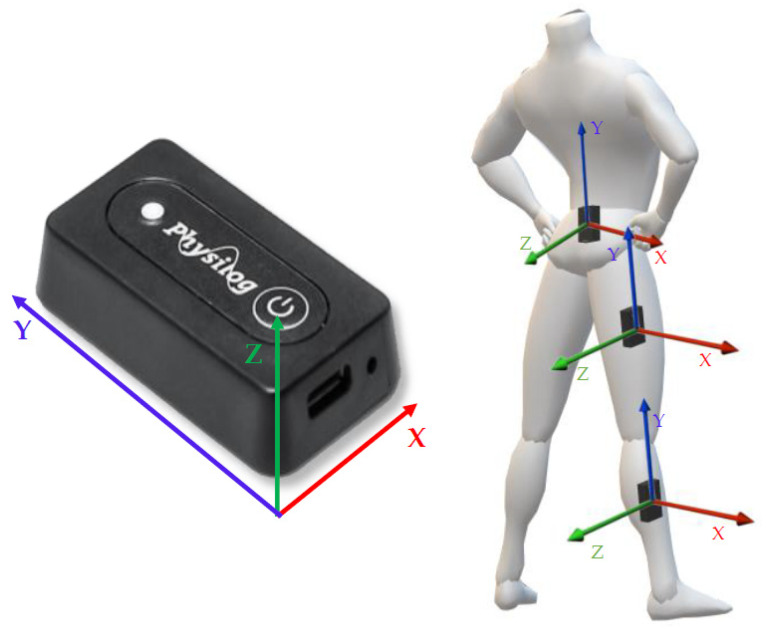
Visual representation of the Physilog sensor’s reference frame (**left**) and representation of the locations of the sensors with their reference frame (**right**).

**Figure 3 sensors-21-07709-f003:**
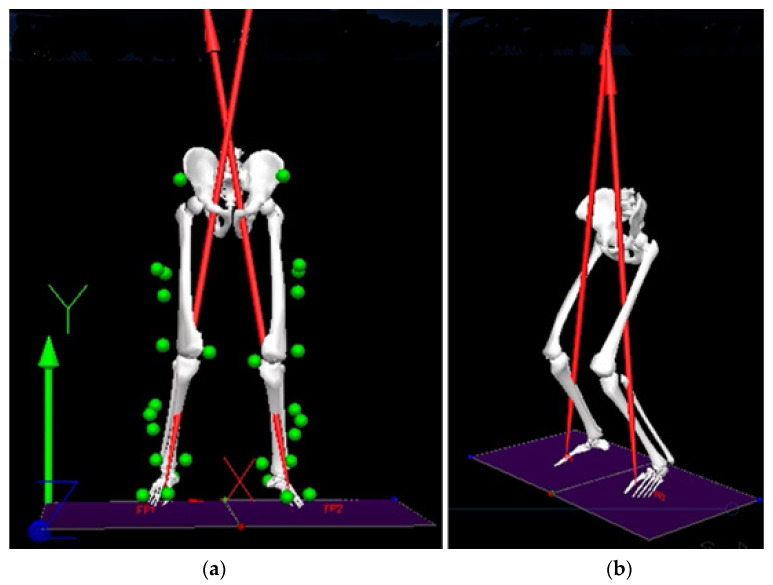
First landing phase of the Vertical Drop Jump in Visual3D, frontal (**a**) and lateral (**b**) views. The green spheres represent the markers on the anatomical landmarks whilst the red arrows are the vertical GRFs.

**Figure 4 sensors-21-07709-f004:**
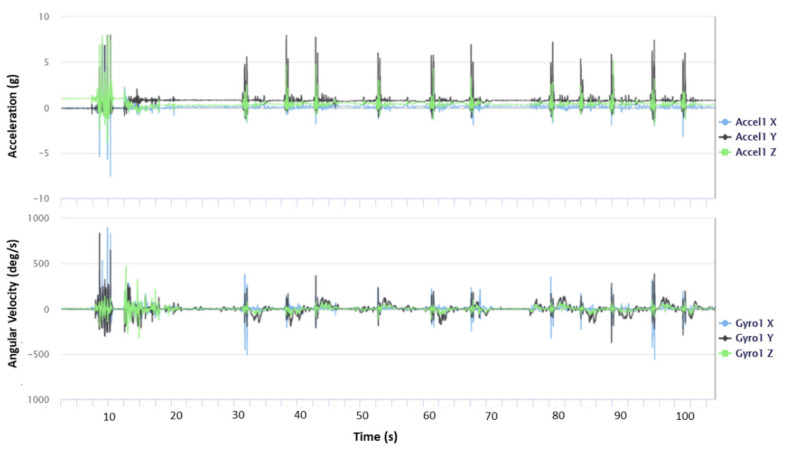
Signals recorded by the IMU placed on the sacrum, displayed in the RKS. The signals are the 3D accelerations (**top**) and the 3D angular velocities (**bottom**).

**Figure 5 sensors-21-07709-f005:**
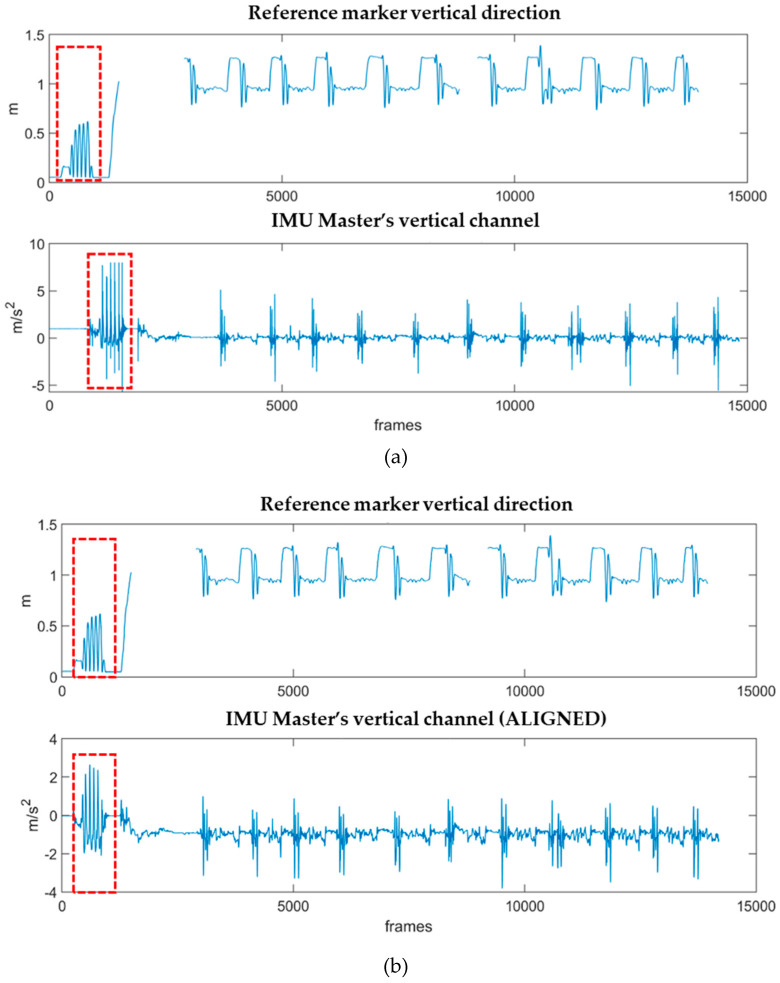
(**a**) Vertical position of the reference marker along the *y*-axis of the laboratory (vertical, positive upwards) and acceleration data along the *z*-axis of the master sensor, attached to the reference marker (vertical, positive downwards). The spikes used as synchronization trigger are visible at the very beginning of each signal (red square); (**b**) Aligned signals of reference marker and acceleration of the master by using the delay information coming from the cross-correlation calculation.

**Figure 6 sensors-21-07709-f006:**
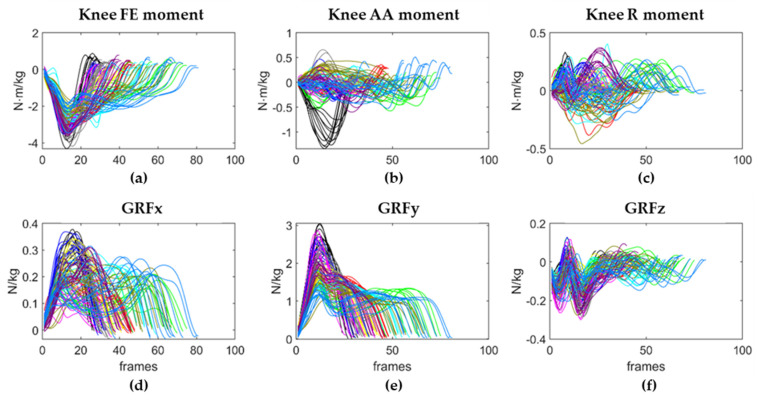
Right knee joint moments (**a**–**c**) and right GFRs (**d**–**f**) during the 112 VDJs. The number of frames that composes each VDJ is highly variable, depending on the jumping behavior of the subjects. Each color corresponds to the VDJs performed by a subject.

**Figure 7 sensors-21-07709-f007:**
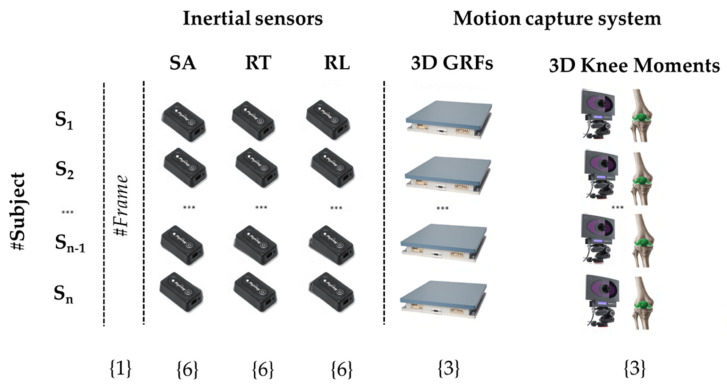
Visual structure of the dataset. The first column contains the reference index of the frame of each jump, columns 2–19 contain the data recorded by the three IMUs, placed on the sacrum (SA), right leg (RL), and right thigh (RT), whilst columns 20–25 contain the 3D right GRFs and the right knee moments. Numbers in parentheses indicate how many features (columns) belong to each source.

**Figure 8 sensors-21-07709-f008:**
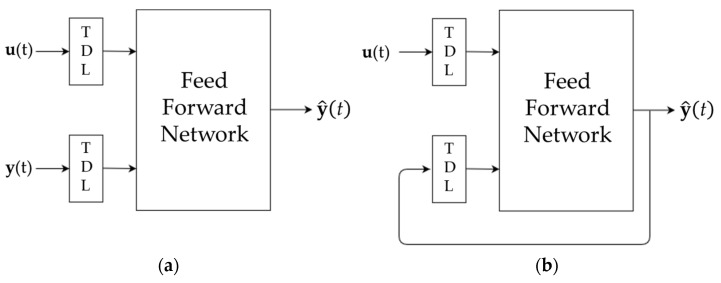
Architectures of the NARX neural network. (**a**) Series–Parallel. (**b**) Parallel.

**Figure 9 sensors-21-07709-f009:**
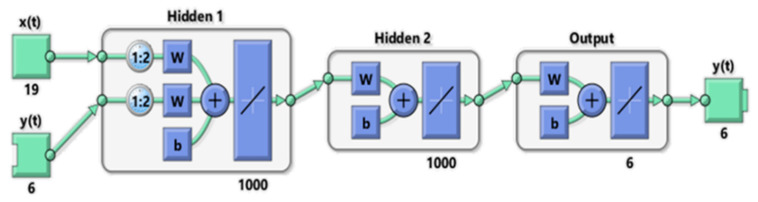
NARX Neural Network topology.

**Figure 10 sensors-21-07709-f010:**
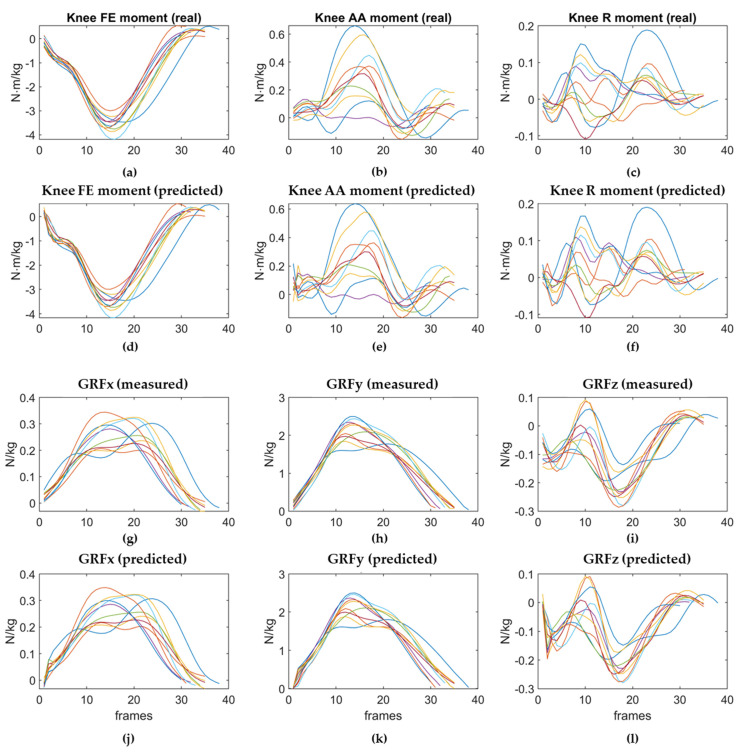
Real and predicted kinetic curves of subject 8. The predicted variables are the the Flexion/Extension (FE) moment, the Adduction/Abduction (AA) moment, and the Rotation (R) moment at the right knee joint, and the 3D GRFs.

**Figure 11 sensors-21-07709-f011:**
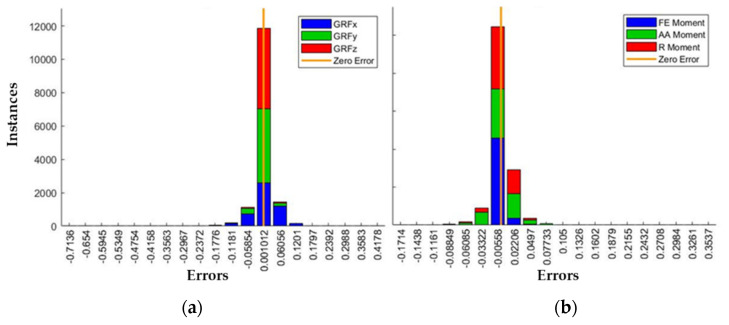
Error histograms of the whole test set. (**a**) Histogram of the errors on the prediction of the right Ground Reaction Forces; (**b**) Histogram of the errors on the prediction of the moments at the right knee joint.

**Figure 12 sensors-21-07709-f012:**
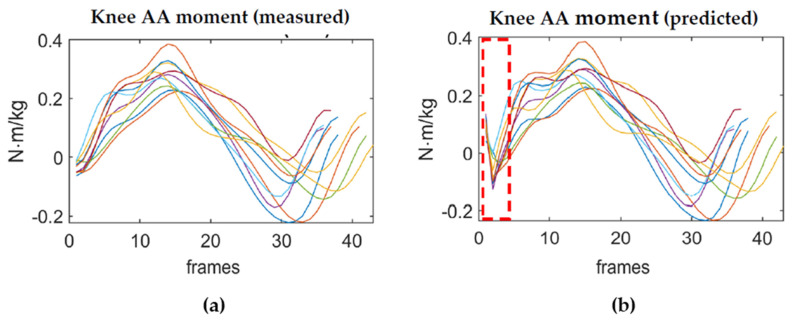
Adduction/abduction moments. (**a**) Measured right knee AA moment; (**b**) Predicted right knee AA moment. Each curve corresponds to a jump performed by a subject and as it can be noticed, the prediction errors accumulated at the beginning of the curves, in the part enclosed by the dotted rectangle.

**Table 1 sensors-21-07709-t001:** Data and anthropometric characteristics of the participants.

Subject ID	Age (Years)	Gender	Height (cm)	Weight (kg)
1	29	M	180	77.5
2	24	M	178	80.0
3	24	F	165	61.8
4	24	F	150	50.8
5	26	F	170	57.4
6	24	M	181	60.1
7	23	M	173	84.3
8	23	M	180	65.9
9	24	F	161	58.0
10	24	F	160	51.5
11	25	M	178	67.8
Mean	24.5		170.5	65.0
SD	1.7		10.3	11.3

**Table 2 sensors-21-07709-t002:** Conventions used to represent knee moments.

Axis	Movement	Positive	Negative	Label
*x*	Flexion/Extension	Flexion	Extension	FE
*y*	Adduction/Abduction	Varus	Valgus	AA
*z*	Rotation	Internal	External	R

**Table 3 sensors-21-07709-t003:** Signals fed to the network (inputs or predictors) and outputs (or targets).

Predictor/Target #	Real Data
Predictor 1	Frame reference indexes
Predictor 2	Angular velocity *x* (gyroscope, SA)
Predictor 3	Angular velocity *y* (gyroscope, SA)
Predictor 4	Angular velocity *z* (gyroscope, SA)
Predictor 5	Acceleration *x* (accelerometer, SA)
Predictor 6	Acceleration *y* (accelerometer, SA)
Predictor 7	Acceleration *z* (accelerometer, SA)
Predictor 8	Angular velocity *x* (gyroscope, RT)
Predictor 9	Angular velocity *y* (gyroscope, RT)
Predictor 10	Angular velocity *z* (gyroscope, RT)
Predictor 11	Acceleration *x* (accelerometer, RT)
Predictor 12	Acceleration *y* (accelerometer, RT)
Predictor 13	Acceleration *z* (accelerometer, RT)
Predictor 14	Angular velocity *x* (gyroscope, RL)
Predictor 15	Angular velocity *y* (gyroscope, RL)
Predictor 16	Angular velocity *z* (gyroscope, RL)
Predictor 17	Acceleration *x* (accelerometer, RL)
Predictor 18	Acceleration *y* (accelerometer, RL)
Predictor 19	Acceleration *z* (accelerometer, RL)
Target 1	Right Knee Flexion/Extension (FE) Moment
Target 2	Right Knee Adduction/Abduction (AA) Moment
Target 3	Right Knee Rotation (R) Moment
Target 4	Ground Reaction Force x
Target 5	Ground Reaction Force y
Target 6	Ground Reaction Force z

**Table 4 sensors-21-07709-t004:** Mean, maximum, and minimum RMSE values corresponding to each target variable predicted by the network according to cut-off frequency of the Butterworth Low-pass filter applied to IMU data. The predicted variables are the 3D GRFs, the Flexion/Extension (FE) moment, the Adduction/Abduction (AA) moment, and the Rotation (R) moment at the right knee joint.

	RMSE																	
	Ground Reaction Forces (N/kg)									Right Knee Moments (N·m/kg)								
Cut-Off Frequency	GRFx			GRFy			GRFz			FE			AA			R		
	m	max	min	m	max	min	m	max	min	m	max	min	m	max	min	m	max	min
12 Hz	0.0290	0.0381	0.0192	0.0433	0.0606	0.0232	0.0409	0.0587	0.0209	0.1159	0.2161	0.0982	0.0302	0.0465	0.0220	0.0114	0.0196	0.0090
24 Hz	0.0087	0.0107	0.0031	0.0381	0.0644	0.0103	0.0188	0.0389	0.0140	0.0816	0.1171	0.0692	0.0334	0.0593	0.0292	0.0121	0.0498	0.0223
32 Hz	0.0081	0.0102	0.0072	0.0350	0.0460	0.0338	0.0184	0.0198	0.0152	0.0748	0.0776	0.0725	0.0295	0.0348	0.0248	0.0117	0.0130	0.0106

## Data Availability

A raw dataset is available online at the following repository: “VDJ_NNdataset”, Mendeley Data, V1, doi:10.17632/8hsrymnzb6.1.

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
