# Peer review of "Machine Learning-Based Estimation of Ground Reaction Forces and Knee Joint Kinetics from Inertial Sensors While Performing a Vertical Drop Jump"

_sensors, 2021, doi:10.3390/s21227709_

Round 1

Author Response

In the following, we provide a description of the changes introduced, which are also highlighted in the manuscript.

We would like to thank you and the Reviewers for their positive comments as well as for the time and expertise they devoted to improving the quality of our MS. We hope that this revised version will be suitable for publication on Sensors.

#REVIEWER 1

Major comments

R1.A: Introduction: While generally well written, the authors gloss over the fact that there is in fact very limited evidence to suggest that the VDJ (or indeed any clinical assessment) can predict future MSK injuries. I believe this is one of the strengths of IMU instrumented assessments, as they provide us with the opportunity to collect objective biomechanical data which could not previously be collected outside of laboratory environments. This approach may allow us to now highlight the link between injury risk and these assessments. However, this needs to be acknowledged throughout the introduction and manuscript as a whole, because current evidence does not necessarily tell us that (yet?).

We totally agree with the Reviewer. There’s still no evidence pointing towards a clear link between injury risk and instrumental data. This might be due to insufficient / not capillary testing. This was now highlighted in the text, with proper references (Kristianslund et al., 2013; Smith et al., 2012; Cronström et al., 2020; Lucarno et al., 2020; Hewett et al., 2005; Hewett et al., 2016); in particular, we changed the Introduction section and the last rows of the discussion.

R1.B Methods: My biggest concern with this work is related to the small sample recruited and the corresponding use of the hold-out set. With this type of data it is normal to have a high degree of inter-subject variability. This is clearly highlighted in Figure 6: Knee AA moment where one subject (black line) is very clearly different to the majority of the other subjects. With this in mind, it may be more appropriate to use a leave-one-subject-out approach to cross-validation here. This way, at least we get a representative idea of the error across the entire dataset, and how sensitive this approach is to new unseen data. For example, I would expect that the subject with the black line would have a high degree of error when held out on this occasion and this should be reflected in your results. To address this, please complete the evaluation using the leave-one-subject-out cross validation and report the results using this method. This will greatly strengthen your results and reduce any bias introduced by the partition used in this small data-set.

We thank the Reviewer for the opportunity to clarify this point. As we were obviously aware of the considerable between-subjects variability (which ultimately adds to the consistency of our study), what we did was a 20-fold cross-validation procedure. We randomly chosen for 20 times 7 out of the 11 subjects as the training set, and 4 out of 11 subjects as the test set. The performance values provided are the average of the 20 iterations. This point was now clarified in the text, lines 383 and following.

R1.C Results: Please provide the range for the RMSE error in table 4 to provide more transparency here. In addition, as the clinical interpretation of this data typically involves looking at the minimum and maximums of the moment of GRF data, it would be really useful if you could provide measures of RMSE error for the minimums and maximums of the predicted time series. For example, how different was the predicted max GRFy from the measured GRFy. This will help provide some clinical context and highlight the error in the typically used clinical measure.

We now added maximum and minimum values of the obtained RMSE to Table 4. We thank the Reviewer also for the following insight. We are aware that discrete parameters could still represent a handy way of summarizing curves, while this paradigm in biomechanics is currently changing to a more holistic approach. While there’s no doubt that this would add some value to the clinical interpretation of results (which by the way could not be the main scope of Sensors), this further analysis goes would require considerable additional effort and goes beyond the constraints imposed by the Journal’s Editor, so we are open to new analyses but we wait for an explicit request from the Editorial board.

Minor comments

R1.1 Abstract L20: Please specify the ‘train-test’ technique used in the abstract

The technique is now specified in the text as ‘train-test split’.

R1.2 L49: I think it should be acknowledged that IMUS don’t necessarily solve all of the issues around sensor placement and skin movements artefacts.

The Reviewer is right, IMUs don’t solve all of the issues around placement and artefacts. An additional paragraph was added to reinforce this aspect, with proper references (page 2, line 65).

R1.3 L65: ‘to with’ should be changed ‘to which’

Changed in the text, thank you for noticing.

R1.4 L86-91: Firstly, this is a very long sentence which may confuse the reader, so I would consider rewording. Secondly, I think the second half of it needs to be toned down. It is a big statement to state relevant kinematic and kinetic data can be determined from a wearable system as many of the studies which have attempted to do this have done either a poor job, or the model developed is so idealistic, it does not deal well with ‘real-world’ unseen data. While a binary classification task, I think this study (https://doi.org/10.3390/s21072346) highlights this problem well.

We shortened and split this sentence for better readability. We also thank the Reviewer for suggesting that interesting reference. We now reworded that sentence so that the actual scope and boundaries of IMUs-based motion capture are now fulfilled.

R1.5 L103-105: please move the makes of the sensors, motion capture and force plate to this sentence as they are currently spread throughout the methods.

Now the producers of the sensors are mentioned in the aforementioned sentence as required

R1.6 L107-112: what was the athletic capacity of the cohort? Where they athletes, did they partake in physical activity and if so, at what level. This is important to share as it will have implications for the variance in the data and also the potential generalisation of this data to athletic cohorts.

Of the considered cohort, three subjects were regular amateur sports practitioners, whilst the other were students with an active lifestyle, occasionally attending the gym and used to walk for daily commuting. This information was added to the text as required (page 3, line 115). As a matter of fact, the inter-individual variability in terms of physical fitness and lifestyle habits is going to positively affect the generalizability of results, embracing a varied population sample.

R1.7 L252-258: Please provide references or justifications for the filter parameters used within this study.

The cut off frequency of 10 Hz (zero-lag Butterworth filter) fits most biomechanics’ applications range(https://doi.org/10.3390/s21134580). This choice was however used just for the devices synchronization.

R1.8 L160-163: please provide specifics of sensor locations and also the orientation of the sensor in relation to this location. Perhaps Figure 2 could be replaced to illustrate the location & the orientation together.

Figure 2 was redrawn from scratch, as suggested, to show all the requested information. The specifics or sensors’ locations were also reported in the text (lines 190-192).

R1.9 Figure 11: Is this the histogram for every data point? If so, please clearly highlight this in the caption

Thank you for this note. Yes, the histogram collects all the errors evaluated on the test set.

R1.10 Line 483-486: how did you quantify this 90% figure? Also, please see my major comment related to the peak of the varo/varus moment. Please address this and provide a measure of error for

appraisal.

We changed this somewhat arbitrary 90% figure into more qualitative terms. We just intended to say that, besides boundary errors, the remainder of the estimated curve fit the corresponding real data.

R1.11 Conclusions: please include an acknowledgement here related to the relatively small size of the dataset and resultant need for more generalisable research.

As suggested, we acknowledged such items in the Conclusion section.

Reviewer 2 Report

Knee loading studies indicated in line 92 should be referenced.
Figure 5 is not referenced in the text.
Looking at the horizontal scale of Figures 5 and 6, they do not appear to be sampled at the same frequency (128 hz) as indicated in the text.
The systematic trial-and-error optimization process carried out to determine the ANN topology (line 309) should be described, at least in summary, since the results obtained depend directly on that process.
There is an error in the units used for the knee moments in Table 4.

Author Response

In the following, we provide a description of the changes introduced, which are also highlighted in the manuscript.

We would like to thank you and the Reviewers for their positive comments as well as for the time and expertise they devoted to improving the quality of our MS. We hope that this revised version will be suitable for publication on Sensors.

# Reviewer 2:

We want to thank the Reviewer for his/her positive comments and suggestions that ultimately helped us in improving the quality of our manuscript.

R2.1 Knee loading studies indicated in line 92 should be referenced.

The aforementioned studies were properly references as suggested (currently line 95).

R2.2 Figure 5 is not referenced in the text.

Now Figure 5 was correctly cited in the text (page 7, line 231).

R2.3 There is an error in the units used for the knee moments in Table 4

The units are now correct.

R2.4 Looking at the horizontal scale of Figures 5 and 6, they do not appear to be sampled at the same frequency (128 hz) as indicated in the text.

The VDJ performed by a subject were performed without stopping the recording between a jump and another. In Figure 5 is displayed the full signal acquired during a session, including the gap between a jump and another, whilst in figure 6 you can see the signals only during the landing phase of each performed VDJ. The sampling frequency is the same, what changes is the portion of signal that is displayed. The horizontal scale is however homogeneous, by “frames” we intended the recorded samples.

R2.4 The systematic trial-and-error optimization process carried out to determine the ANN topology (line 309) should be described, at least in summary, since the results obtained depend directly on that process.

We thank the Reviewer for giving us the chance of further explaining this point. What we’ve done to determine the final ANN structure was to start from a set of hyperparameters (i.e., number of hidden layers, number of neurons therein, and so on) obtained from similar existing research (e.g., Mundt et al., 2019). Then, such parameters were adjusted iteratively based on the network’s output metrics. This was explained in Materials and Methods, lines 323-329.

Reviewer 3 Report

1. the configuration of the IMUs is also expected to be showed in Fig.1 along as the markers, which will be more clear and understandable. The "tight" in line 161 should be "thigh". 2. Is the 128 Hz sampling frequency fast enough for catching the human state at the moment of landing? The Fig.4 showed that the maximum acc exceeded the measurement range the IMU ? 3. Remove the red line in Fig. 8. 4. There are a few typos that should be corrected.

Author Response

In the following, we provide a description of the changes introduced, which are also highlighted in the manuscript.

We would like to thank you and the Reviewers for their positive comments as well as for the time and expertise they devoted to improving the quality of our MS. We hope that this revised version will be suitable for publication on Sensors.

****

#Reviewer 3

R3.1 the configuration of the IMUs is also expected to be showed in Fig.1 along as the markers, which will be more clear and understandable.

Thank you for this suggestion. We considered this solution in the first draft of the paper. However, it was decided to stick to the actual version as it was unfeasible to draw a figure showing all the markers and the placements of the sensors clearly. This is why the configuration of the IMUs was added in Figure 2, also following the advice of Reviewer 1.

R3.2 The "tight" in line 161 should be "thigh".

Changed in the text, thank you for noticing.

R3.3 Is the 128 Hz sampling frequency fast enough for catching the human state at the moment of landing?

We believe that such sampling frequency (period: ~8 ms) was appropriate to catch the jumping action, which lasts approximately 300 ms. This choice was also selected as:

  1. a) 128 Hz was the nearest frequency with respect to the optoelectronic system sampling freq. (100 Hz) - since to assemble the machine learning system that followed, we had to sync the two systems.
  2. b) it was the highest frequency allowed for three IMU sensors to be synched together.

R3.4 The Fig.4 showed that the maximum acc exceeded the measurement range the IMU?

Thank you for this note. Actually, while working at this research, we noticed the maximum accelerations sometimes saturated the dynamic range, and then we set such limits to +/- 16 g, but we did not change the original picture. Fig. 4 now describes a sequence of jumps which do no exceed the range.

R3.5 Remove the red line in Fig. 8.

The figure was sorted accordingly to the aforementioned advice, thank you for noticing.

R3.6 There are a few typos that should be corrected.

The whole text was carefully checked.

Reviewer 4 Report

Following the review of the paper, I consider that this is an excellent research that presents a new study that investigates the use of an ANN structure to predict knee joint kinetics.

Although the authors mention that they present a detailed description of the NARX architecture, they can do more to achieve this goal. If a third party wants to implement this architecture based on the same datasets (congratulations to the authors for the idea to make this dataset public) in order to verify the performances reported in the paper, certainty it will not be able to do it.  Details such as: (A) regularization approach {“regularization was introduced to prevent overfitting” (316)} or activation functions { “different activation functions for each layer” (320 & 321)} are not presented in the paper.

Another aspect that I think should be clarified: is there any motivation based on which the cutoff frequencies of the low pass filters were chosen {“with a 32 Hz cut frequency” (254) & “cut frequency of 12 Hz and 24 Hz respectively to evaluate the sensitivity of the outcome to the filter settings” (267 & 268)}?

I also think that the following sentence should be reworded: “Processed data were to Matlab with custom scripts” (209 & 210).

Author Response

In the following, we provide a description of the changes introduced, which are also highlighted in the manuscript.

We would like to thank you and the Reviewers for their positive comments as well as for the time and expertise they devoted to improving the quality of our MS. We hope that this revised version will be suitable for publication on Sensors.

***

#Reviewer 4

R4.1 Following the review of the paper, I consider that this is an excellent research that presents a new study that investigates the use of an ANN structure to predict knee joint kinetics.

We thank the Reviewer for his/her appreciation.

R4.2 Although the authors mention that they present a detailed description of the NARX architecture, they can do more to achieve this goal. If a third party wants to implement this architecture based on the same datasets (congratulations to the authors for the idea to make this dataset public) in order to verify the performances reported in the paper, certainty it will not be able to do it.  Details such as: (A) regularization approach {“regularization was introduced to prevent overfitting” (316)} or activation functions { “different activation functions for each layer” (320 & 321)} are not presented in the paper.

We thank the Reviewer for the chance to further detail our research. We now added some numerical figures about regularization (as the factor equal to 0.5, [line 364]).

We also acknowledge that we could have added details concerning the activation functions: we specified the activation functions tested in M&M; in the Results section we explicitly mentioned the “purelin” linear transfer function, which was ultimately used.

R4.3 Another aspect that I think should be clarified: is there any motivation based on which the cutoff frequencies of the low pass filters were chosen {“with a 32 Hz cut frequency” (254) & “cut frequency of 12 Hz and 24 Hz respectively to evaluate the sensitivity of the outcome to the filter settings” (267 & 268)}?

At first, we choose the cut frequency as the frequency at the 90% of the PSD in preliminary data. Then we scaled this cutoff frequency, as mentioned in the test, with the aim of evaluating which was the sensitivity of the model to different data conditioning choices.

If the Reviewer thinks this might be misleading, we could also consider to keep only the 32 Hz filtering.

R4.4 I also think that the following sentence should be reworded: “Processed data were to Matlab with custom scripts” (209 & 210).

The Reviewer is correct, thank you for noticing.

The sentence was reworded in the text (now line 228-229).

Round 2

Reviewer 3 Report

The authors have addressed these questions.